# Toward Real Ultra Image Segmentation: Leveraging Surrounding Context to Cultivate General Segmentation Model

**Sai Wang, Yutian Lin, Yu Wu,* Bo Du**
School of Computer Science, Wuhan University
{wangsai23, wuyucs, yutian.lin, dubo}@whu.edu.cn

## Abstract

Existing ultra image segmentation methods suffer from two major challenges, namely the scalability issue (i.e. they lack the stability and generality of standard segmentation models, as they are tailored to specific datasets), and the architectural issue (i.e. they are incompatible with real-world ultra image scenes, as they compromise between image size and computing resources). To tackle these issues, we revisit the classic sliding inference framework, upon which we propose a Surrounding Guided Segmentation framework (SGNet) for ultra image segmentation. The SGNet leverages a larger area around each image patch to refine the general segmentation results of local patches. Specifically, we propose a surrounding context integration module to absorb surrounding context information and extract specific features that are beneficial to local patches. Note that, SGNet can be seamlessly integrated to any general segmentation model. Extensive experiments on five datasets demonstrate that SGNet achieves competitive performance and consistent improvements across a variety of general segmentation models, surpassing the traditional ultra image segmentation methods by a large margin.

## 1 Introduction

With the rapid development of computing and imaging equipment, ultra-high resolution images with millions or even billions of pixels emerge in endlessly and driving the need for more advanced analytical techniques. Accurate understanding of information conveyed by images [1, 47, 43] has become imperative in diverse fields such as remote sensing [29, 48, 44, 50] and medical analysis [37, 36, 31], and ultra image segmentation has emerged as an essential tool for achieving this goal.

To achieve ultra image segmentation, most of the methods adhere to the concept of integrating global and local information, utilizing the global context clues to aid local region refinement [6, 20]. Typically, these networks have two branches: one receives the down-sampled ultra image and extracts global context information of whole image, while the other extracts local information from sliced patches or the complete image, as shown in Fig. 1(a). The final result is generated by fusing the global and local features. Despite numerous improvements made to ultra image segmentation, we observe that it still suffer from issues in both their scalability and architecture:

(1) **Scalability Issue.** The existing ultra image segmentation methods (UIS) often rely on dataset-specific parameters, which limit their model capacity and scalability. The UIS methods struggle to scale up to larger image sizes, with performance significantly degrading as image resolution increases. Moreover, UIS has complex training procedures that need multi-stage parameter tuning [8, 24]. In contrast, general semantic segmentation (GSS) methods are more effective and demonstrate greater

---

*Corresponding author

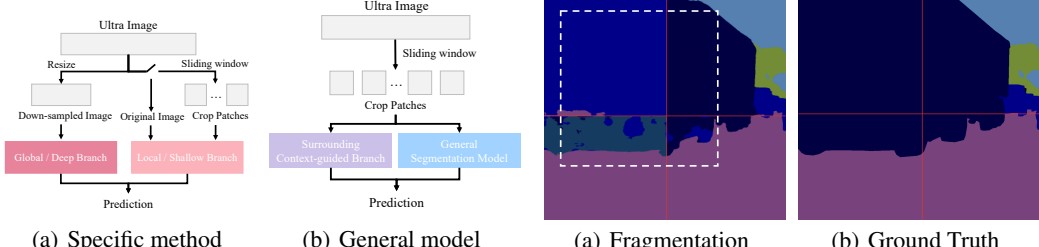

(a) Specific method     (b) General model     (a) Fragmentation     (b) Ground Truth

Figure 1: Comparison with specific (a) and (b). The previous architecture all suffered from **Scalability Issue** and **Architectural Issue** due to specific designs and a lack of tailored context, whereas ours is designed to leverage any general segmentation model to address ultra image segmentation.

Figure 2: Directly adapting the general segmentation model to ultra image scene will cause **fragmentation phenomenon (a)**: the prediction results of the edge areas between adjacent patches (even in overlapped patches) are inconsistent.

scalability compared to UIS. With a straightforward training process, GSS can be adaptable to various ultra image datasets.

(2) **Architectural Issue.** The architecture of existing UIS is not suitable for processing ultra image scenes in real-world scenarios, as it compromises between dataset size and computing resources. Most methods utilize either the entire image or the downsampled version to capture global information. However, the two strategies are both constrained by the size of input image. When the input image is excessively large, the former strategy cannot process it directly, while the later will suffer great information loss during compression.

Building upon the above discussions, a simple solution is to introduce general segmentation model [27, 12, 18, 39] into the ultra image segmentation task using the sliding window approach, which takes only isolated patches as input due to memory limitations. However, directly adapting GSS to ultra image scenes also raises two challenges: (1) **Information bottleneck.** Using isolated patches as input prevents the model from capturing the correlations between patches, which blocks the information flow and affects the model's perception of the surrounding information. (2) **Fragmentation phenomenon.** With isolated patch input, even if overlapped patches are used, the prediction results of each patch are independent. This leads to inconsistent prediction between patches, especially at the edges of adjacent patches, as shown in Fig. 2.

Based on above considerations, we propose an end-to-end framework called S̲urrounding G̲uided Segmentation Framework (SGNet), which takes advantage of the general segmentation method and utilizes surrounding information near the local patch to guide the model. As shown in Fig. 1(b), SGNet includes two decoupled parts: a general segmentation module and a surrounding context-guided branch. Specifically, SGNet takes both the local patch and the corresponding surrounding patch as input. The surrounding patch covers a larger area around the local patch and provides more context information. In the surrounding context-guided branch, we model the context information required to segment the local patch from a larger perspective to drive the flow of information between regions of whole image. Besides, to alleviate the fragmentation phenomenon, we propose a boundary consistency loss to improve the consistency of the prediction results of adjacent patches, thereby alleviating the inconsistent predictions. Note that, unlike existing tightly coupled ultra images segmentation methods, our method can be seamlessly incorporated into any general segmentation models, and brings stable performance improvements.

Our contributions are summarized as follows:

- We excavate two essential but largely overlooked issues in UIS, which hold great value for the community. In addressing these challenges, we are the first to tackle the ultra image segmentation task from the general segmentation model perspective.

- We present a novel end-to-end ultra image segmentation framework named SGNet, which leverage surrounding context information to guide patch-based model segmentation. Our method is flexible to be added to any general segmentation model.

- Experiments show that our method achieves competitive performance and consistently improves over different general segmentation models on five public datasets, outperforming previous methods by a large margin.

## 2 Related Work

### 2.1 General Semantic Segmentation

With the development of deep learning [33, 32], the two mainstream methods based on convolution neural network [52, 35, 34, 2] and Transformer [42, 17, 51, 28] have achieved excellent performance. FCN [27] is the first fully convolutional architecture and a lot of work has been extended, such as DeeplabV3 [3] and HRNet [38]. Compared with CNN-based methods, representative Transformer-based works include SegFormer [46], Mask2Foremr [7] and SAM [22]. BiSeNetV1 [49] and STDC [12] are designed for real-time segmentation to reduce the computational overhead. The general segmentation model shows excellent stability and scalability, and we aim to leverage its strengths to address both the Scalability and Architectural issues in existing UIS methods.

### 2.2 Ultra Image Semantic Segmentation

GLNet [6] introduces a novel global-local architecture. PPN [45] builds on the top of GLNet by integrating a classification network to distinguish valuable patches. Furthermore, Magnet [16] employs a multi-stage pipeline where each stage corresponds to a specific magnification level. FCtL [24] exploits locality-aware contextual correlation to effectively integrate and associate contextual information of local patches. Compared to previous methods, ISDNet [14] proposes a novel framework that combines shallow and deep networks, enabling directly whole-image inference, thereby improving the segmentation effect while increasing the speed. ElegantSeg [5] proposes a end-to-end holistic learning framework from the perspective of engineering optimization. Recently, WSDNet [20] follows the architecture of ISDNet, using DWT-IWT to preserve spatial details. GPWFormer [19] also employs the global-local architecture, and propose the wavelet transformer to model semantic relations. However, the aforementioned methods all suffer from inherent framework flaws, and cannot be applied to ultra images of extremely large scale. On the contrary, our simple yet generalized solution is more adaptable and applicable than specific UIS methods in real-world scenarios.

## 3 Methodology

### 3.1 Overview

In this section, we present SGNet, a novel framework that enables existing general segmentation models for ultra image segmentation. As shown in Fig. 3, SGNet consists of two major components, the general segmentation module and surrounding context-guided branch (SCB). The two branches take the local patch and its surrounding larger area as inputs, respectively and extract their features (Section **Architecture**). In the surrounding context-guided branch, we introduce a surrounding context integration module (Section **Surrounding Context Integration Module**) to enable interaction between local and surrounding features, and selectively learn contextual information that is helpful for patch segmentation. Furthermore, we propose a boundary consistency loss to maintain the consistency of prediction results across adjacent patches in Section **Loss Function**.

### 3.2 Architecture

The general segmentation approaches include global inference and slide inference. The former approaches could result in a significant quality drop during the image resolution compression. Therefore, we attempt to adapt the slide inference based methods into ultra image segmentation tasks. Given an ultra image $I \in \mathbb{R}^{H \times W \times C}$, we divide it into N non-overlapping local patches $I_{local} \in \mathbb{R}^{h \times w \times c}$ and feed them into general segmentation module for prediction as our target. Next, we start from a random position within the local patch as the center and obtain a surrounding patch that is $\alpha(\alpha > 1)$ times larger than its side length. The surrounding patch includes more contextual details, which is helpful to guide the local patch training.

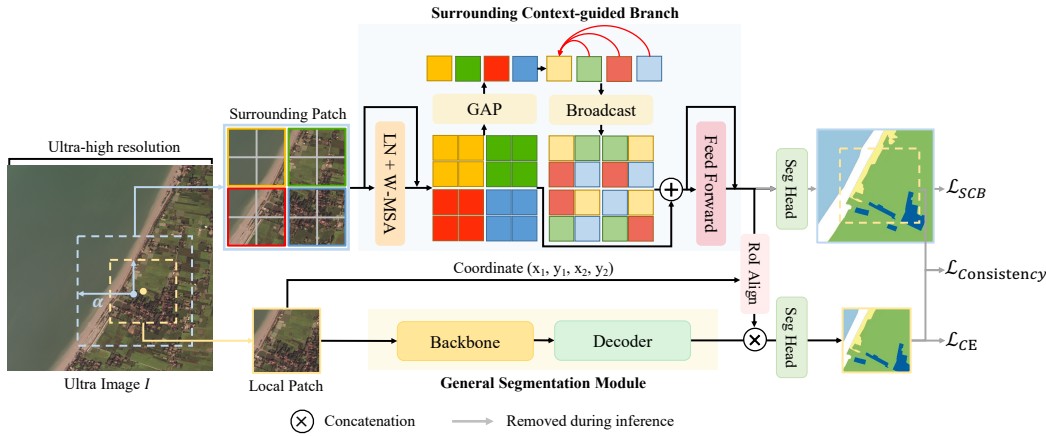

Figure 3: The Architecture of SGNet. An ultra image $I$ is randomly cropped to obtain a local patch $I_{local}$ and a surrounding patch $I_{global}$ containing more context information of any size, which are respectively sent to the general segmentation module and the surrounding context-guided branch to extract features. The resulting features are aggregated through simple concatenation and used to generate high-quality predictions. General segmentation module can be applied to any segmentation model, and surrounding context-guided branch consistently achieves stable improvements on it. LN, W-MSA, and GAP stand for layer normalization, window-based multi-head self-attention, and global average pooling, respectively.

Afterward, the general segmentation module receives the local patch to extract features. Meanwhile, the surrounding context-guided branch (SCB) receives the surrounding patch to extract the surrounding information to guide the segmentation of local patch. In order to boost the processing speed of the SCB, a lightweight backbone is employed for feature extraction. The resulting feature map are subsequently transmitted to the surrounding context integration module for further relationship modeling and focused acquisition of contextual information essential for local patch prediction.

The output feature map is aligned using coordinate relative relationships, retaining only the portion corresponding to the local patch. This portion is then combined with the feature map generated by the general segmentation module to obtain the final prediction through a standard segmentation head. To achieve a more complete optimization of the SCB, we add an extra auxiliary segmentation head to predict the result of the surrounding patch. This helps us to further calculate the boundary consistency loss between the local and surrounding patches based on this prediction.

Compared to previous ultra image segmentation methods [6, 45, 24, 14, 19] that cannot handle extremely large images well, our architecture is more flexible and integrated, as it can handle an **ANY** size large image, equip with **ANY** general segmentation model, thereby well-suited for real-world scenarios.

### 3.3 Surrounding Context Integration Module

The information contained in the surrounding patch can serve as an extension of the local patch, providing it with more abundant decision-making guidance. Therefore, we introduce the Surrounding Context Integration Module (SCI), from the perspective of absorbing the information contained in each region, and integrating context across all the windows of surrounding patch. The structure of SCI is shown in Fig. 3.

Following [26, 25, 13], the feature map $F \in \mathbb{R}^{H \times W \times C}$ is partitioned into a set of $R \times R$ non-overlapping regions as $F_{region} \in \mathbb{R}^{\frac{H}{R} \times \frac{W}{R} \times R^2 \times C}$, with each region being subsequently subdivided into $w \times w$ non-overlapping windows as $F_w \in \mathbb{R}^{\frac{H}{Rw} \times \frac{W}{Rw} \times R^2 \times w^2 \times C}$.

To facilitate information exchange within the region as well as absorb the contextual information that is helpful for segmentation, we perform layer normalization and window-based multi-head self-attention (W-MSA) on all patch tokens within $F_w$ as follows:

$$F_w^{'} = W\text{-}MSA(LN(F_w)) + F_w. \tag{1}$$

Currently, the information of all patch tokens within the region have been fully incorporated into $F_w'$. Subsequently, we apply global average pooling (GAP) to acquire the global feature representation $F_w^{global} \in \mathbb{R}^{\frac{H}{Rw} \times \frac{W}{Rw} \times R^2 \times 1 \times C}$ for each window, so as to facilitate the information exchange among the windows in later stage:

$$F_w^{global} = GAP(F_w').$$ (2)

This process enables the absorption and integration of information from every position within the window into a unified vector representation that encapsulates the overall information of the window.

To leverage the complementarity of information and encourage information sharing among windows within the surrounding patch, we utilize self-attention (SA) mechanism to capture the interdependence among the global feature representations $F_w^{global}$ of each window:

$$F_w^{global'} = SA(F_w^{global}).$$ (3)

Upon completion of the information exchange, the global feature representation of the current window incorporates the information of other windows. Subsequently, the global feature representation $F_w^{global'}$ is added to $F_w'$ by broadcasting, while the information of the remaining windows is transferred to $F_w'$:

$$F_{global} = F_w' + F_w^{global'}.$$ (4)

To enhance the compatibility and generalization for subsequent operation, we apply feed forward layer (FFN) to further refine $F_{global}$:

$$F_{global}' = FFN(F_{global}) + F_{global}.$$ (5)

At this time, $F_{global}'$ integrates the context information of the surrounding patch, which can strengthen the features of the local patch region as a complement and solve the challenge of information bottleneck.

### 3.4 Loss Function

#### 3.4.1 Boundary Consistency Loss

Despite the incorporation of local patch features and surrounding contextual information, there are still inconsistent prediction results between adjacent patches due to the lack of explicit constraints. Therefore, we propose a boundary consistency loss $\mathcal{L}_{Consistency}$ to improve the consistency of prediction results in neighboring regions and alleviate the fragmentation phenomenon. $\mathcal{L}_{Consistency}$ compels the prediction results of both the general segmentation module and SCB to be as similar as possible, thus promoting consistency in results of neighboring regions across different patches. This helps create a smoother transition between predictions of adjacent patches, harmonizing the prediction of the entire ultra image.

Concretely, we crop the predicted mask $P_{global}'$ corresponding to the local patch from the prediction of the surrounding patch $P_{global}$. Then, we apply L1 constraints on both $P_{global}'$ and the prediction result of local patch $P_{local}$ to encourage similarity between them, even when different contexts are utilized. The loss function is defined as follows:

$$\mathcal{L}_{Consistency} = ||P_{global}' - P_{local}||.$$ (6)

#### 3.4.2 Overall Loss

The cross-entropy loss is used for both the general segmentation module ($\mathcal{L}_{CE}$) and SCB ($\mathcal{L}_{SCB}$). The overall loss $\mathcal{L}$ is a weighted sum of all the losses mentioned above:

$$\mathcal{L} = \lambda_1 \mathcal{L}_{CE} + \lambda_2 \mathcal{L}_{SCB} + \lambda_3 \mathcal{L}_{Consistency}.$$ (7)

Table 1: Comparison with baseline on five datasets. The "Mode" column denotes the specific inference modes associated with each method. The "Test size" column relates to the DeepGlobe dataset.

| Method | Mode | Backbone | Test size | DeepGlobe 2448×2448 | FBP 6800×7200 | Aerial Inria 5000×5000 | Gleason 5120×5120 | Cityscapes 2048×1024 |
|---|---|---|---|---|---|---|---|---|
| *Ultra Image Segmentation Methods* | | | | | | | | |
| GLNet [6] | Slide+Whole | R50 | 508 | 71.60 | 42.05 | 71.20 | - | - |
| PPN [45] | Slide+Whole | R50 | 512 | 71.90 | - | - | - | 75.20 |
| MagNet [16] | Slide+Whole | R50 | 508 | 72.96 | 44.20 | - | - | 67.57 |
| FCtL [24] | Slide | VGG16 | 508 | 72.76 | 48.28 | 72.87 | - | - |
| ISDNet [14] | Whole | R18 | 2448 | 73.30 | 21.98 | 74.23 | 59.97 | 76.02 |
| ElegantSeg [5] | Whole | W48 | 2448 | 74.32 | 61.62 | - | - | - |
| WSDNet [20] | Whole | R50 | 2448 | 74.10 | - | 75.20 | - | - |
| GPWFormer [19] | Slide+Whole | R50 | 500 | 75.80 | - | 76.50 | - | 78.10 |
| *General Semantic Segmentation Methods* | | | | | | | | |
| FCN [27] | Slide | R50 | 512 | 72.38 | 59.97 | 80.35 | 52.65 | 72.39 |
| + *SGNet* | Slide | R50 | 512 | 75.28 (+2.90) | 61.85 (+1.88) | 80.81 (+0.46) | 58.46 (+5.81) | 75.84 (+3.45) |
| DeepLabV3Plus [4] | Slide | R50 | 512 | 73.22 | 61.85 | 80.88 | 55.45 | 75.24 |
| + *SGNet* | Slide | R50 | 512 | 75.44 (+2.22) | 63.18 (+1.33) | 81.21 (+0.33) | 61.21 (+5.76) | 76.72 (+1.48) |
| HRNet [38] | Slide | W18 | 512 | 72.87 | 58.55 | 79.17 | 54.31 | 71.20 |
| + *SGNet* | Slide | W18 | 512 | 75.25 (+2.38) | 61.49 (+2.94) | 80.08 (+0.91) | 60.50 (+6.19) | 73.06 (+1.86) |
| SegFormer [46] | Slide | Mit-b0 | 512 | 72.96 | 57.56 | 76.26 | 49.62 | 67.08 |
| + *SGNet* | Slide | Mit-b0 | 512 | 74.65 (+1.69) | 60.35 (+2.79) | 78.80 (+2.54) | 54.86 (+5.24) | 70.42 (+3.34) |
| STDC [12] | Slide | R50 | 512 | 72.59 | 54.38 | 75.20 | 54.51 | 66.24 |
| + *SGNet* | Slide | R50 | 512 | 74.51 (+1.92) | 59.40 (+5.02) | 77.25 (+2.05) | 60.36 (+5.85) | 69.36 (+3.12) |

# 4 Experiments

## 4.1 Datasets

To comprehensively evaluate our method, we conduct experiments on five public ultra image datasets involving general, medical and remote sensing scenarios: Cityscapes [10], DeepGlobe [11], Inria Aerial [30], Five-Billion-Pixels [40], and Gleason [21]. Since we do not have data partition details from [16], we randomly split the Gleason dataset into a training set of 195 images and a testing set of 49 images, and retrained the relevant models. All other datasets followed the official division.

## 4.2 Implementation Details

In both training and testing, the local patch has a size of 512×512 in all datasets, while the surrounding patch is twice as large without any resizing. During sliding inference process, we do not preserve any overlapping regions and the center of local patch and surrounding patch are the same. The first four stages of STDC, used as the lightweight backbone following [14], are initialized with ImageNet weights. For all experiments, we set $\lambda_1 = 1, \lambda_2 = 0.4, \lambda_3 = 0.1$. For the DeepGlobe dataset, the "unknown" category is ignored during training as it is not included in the evaluation [11]. We also exclude "unlabeled" category in the FBP dataset following [5].

We adopt MMSegmentation [9] as our toolbox and use AdamW optimizer, which initial learning rate is set to $2 \times 10^{-4}$. All the models are trained on 4 Tesla V100 GPUs with batch size of 8, except for the Inria Aerial and Gleason dataset, which are trained for 10k iterations while the rest are trained for 30k iterations. Apart from regular operations such as multi-scale training, flipping, and rotating, we do not do any special data augmentation, and the final result does not use any test time augmentation.

## 4.3 Comparison Results

We classify our comparison methods into two groups: general semantic segmentation and ultra image segmentation. To perform a comprehensive evaluation, we select semantic segmentation methods that rely on CNN (FCN, DeepLabV3Plus, HRNet), Transformer (SegFormer), and lightweight architecture (STDC) for comparison. In addition, we verify the performance of classic ultra image segmentation methods, including GLNet, ISDNet, GPWFoermer, *etc*. For fair comparison, we adopt ResNet-50 [15] or its equivalent parameter amount as the backbone for all the methods we compared.

### 4.3.1 General *VS* Ultra Image Segmentation Methods

According to the results presented in Table 1, UIS methods only work well on specific datasets, while GSS methods achieve relatively satisfying performance on all datasets. This confirms the aforementioned *scalability issue*. The existing UIS methods show a noticeable performance decrease when transitioning from handling images in DeepGlobe dataset (2448×2448) to larger ultra images

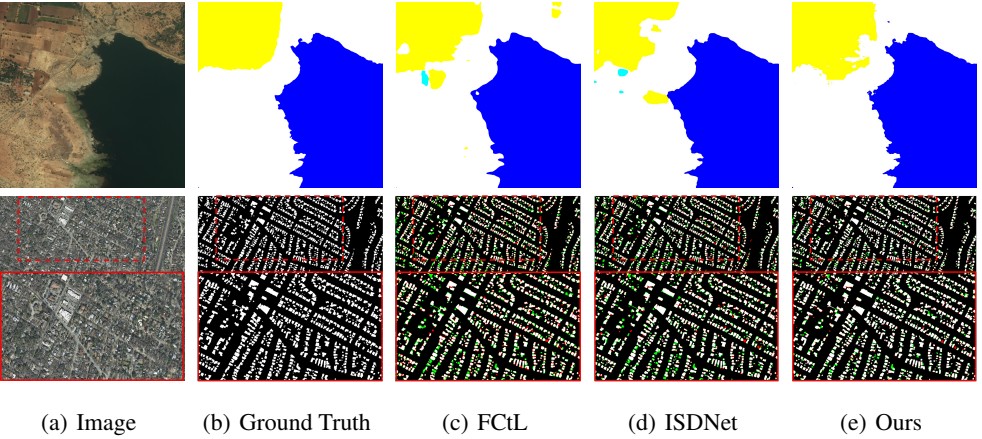

|   (a) Image   |   (b) Ground Truth   |   (c) FCtL   |   (d) ISDNet   |   (e) Ours   |

Figure 4: The visualization of different methods in *DeepGlobe* (top row, 2448×2448) and *Inria Aerial* (bottom row, 5000×5000). We use DeepLabV3Plus as our general segmentation model and add SCB on top of it. We apply red color to mark regions where the background is misclassified as foreground and green color to denote regions where foreground is misclassified as background.

Table 2: Efficacy of proposed module. Align, SCI, Aux, Loss respectively correspond to the feature alignment operation, surrounding context integration module, auxiliary head and boundary consistency loss.

| Group | Backbone | Align | SCI | Aux | Loss | mIoU |
|-------|----------|-------|-----|-----|------|------|
| A | - | - | - | - | - | 73.22 |
| B | ✓ | - | - | - | - | 73.76 |
| C | ✓ | ✓ | - | - | - | 73.88 |
| D | ✓ | ✓ | ✓ | - | - | 74.76 |
| E | ✓ | ✓ | ✓ | ✓ | - | 75.23 |
| F | ✓ | ✓ | ✓ | ✓ | ✓ | 75.44 |

Table 3: Analysis of Surrounding Context Integration.

| Attention | | Global Interaction | | |
|-----------|-------|------|-----|------|
| Naive SA | W-MSA | Conv | GAP | mIoU |
| - | - | - | - | 74.40 |
| ✓ | - | - | - | 74.86 |
| - | ✓ | - | - | 74.72 |
| - | ✓ | ✓ | - | 75.17 |
| - | ✓ | - | ✓ | 75.44 |

like FBP (6800×7200), particularly for those utilizing whole images as input, such as ISDNet. This demonstrates that the current UIS architecture lacks flexibility and faces a trade-off between models and datasets. This reveals the aforementioned *architectural issue*.

### 4.3.2 Improvement over General Segmentation Model

Compared with above methods, our method obtains consistent gains upon all general segmentation methods. Since our method can seamlessly be incorporated into GSS and utilizes the sliding inference strategy for prediction, it can handle ultra images of any scale while leveraging the advantages of scalability from GSS. This perfectly resolves the *scalability issue* and *architectural issue*. Taking the DeepGlobe dataset as an example, our method yields a minimum improvement of 1.69% across all general models, and a maximum of 75.44% mIoU, which is 2.22% higher than using only DeepLabV3Plus. This demonstrates that our method can make full use of the contextual information in the surrounding patch to guide local patch segmentation. It is worth emphasizing that our method is flexible and applicable to any general segmentation models. It also should be noted that all the methods reported in Table 1 are sliding window without overlap. A sliding window with overlap is essentially a test time augmentation method, where multiple predictions on overlapping regions are averaged to enhance the model's performance. By using overlapping regions that cover half of the input patch for predictions, our method can further improve the performance by 0.24% to 75.68% mIoU on DeepGlobe dataset.

Qualitative results of representative works are shown in Fig. 4, while the results of WSDNet and GPWFormer are not available due to no public code. Since we leverage the contextual clues around the current patch as a guide, our method exhibits fewer false positives and delivers more precise segmentation results than other methods.

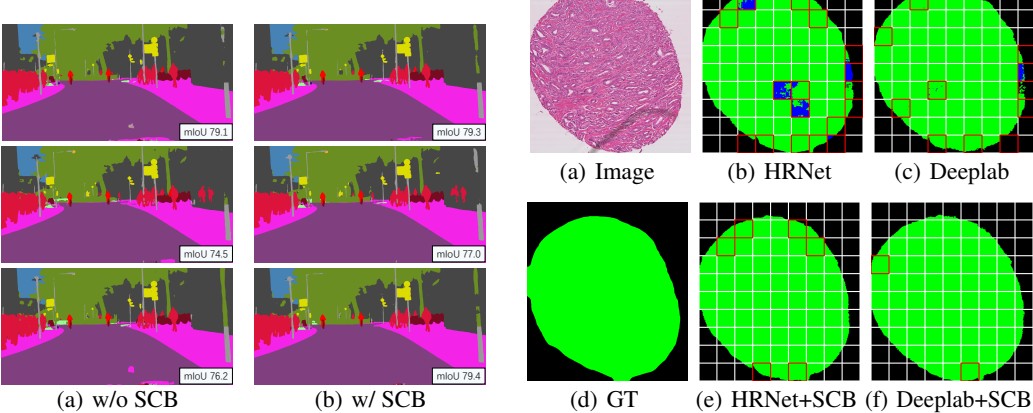

| | |
|---|---|
| (a) w/o SCB     (b) w/ SCB | (d) GT     (e) HRNet+SCB (f) Deeplab+SCB |

Figure 5: Comparison of adding SCB on different models (from top to bottom are DeeplabV3Plus, HRNet and FCN in 5(a) and 5(b)) in *Cityscapes* (2048×1024). The mIoU values are calculated on each shown images.

Figure 6: Comparison of adding SCB on different models in *Gleason* (5120×5120). The predicted patches of model are divided by the white line, while the red box indicates the area where the model failed to predict.

Table 4: Analysis of feature fusion scheme.

| Early Fusion | Late Fusion | ADD | CONCAT | mIoU |
|:---:|:---:|:---:|:---:|:---:|
| ✓ | - | ✓ | - | 74.68 |
| ✓ | - | - | ✓ | 74.87 |
| - | ✓ | ✓ | - | 74.93 |
| - | ✓ | - | ✓ | 75.44 |

Table 5: Analysis of surrounding patch size.

| Scale ($\alpha$) | Surrounding Patch Size | mIoU |
|:---:|:---:|:---:|
| 1 | 512 | 74.75 |
| **2** | **1024** | **75.44** |
| 3 | 1536 | 74.96 |
| 4 | 2048 | 74.78 |

## 4.4 Ablation Study

### 4.4.1 Effectiveness of Proposed Module

To thoroughly confirm the efficacy of each module, we substitute SGNet (DeepLabV3Plus version) with six variations denoted as Group $\mathbb{ABCDEF}$. As shown in Table 2, we employ only a lightweight backbone to individually extract features from surrounding patch, and then concatenated them on last features in group $\mathbb{B}$. Group $\mathbb{B}$ achieves a mIoU of 73.76%, which exceeds group $\mathbb{A}$ that only used general segmentation module by 0.54%. In group $\mathbb{C}$, we utilize the crop operation to align features, resulting in an improvement of 0.12% compared to $\mathbb{B}$. By applying the surrounding context integration module to model the context information, and the auxiliary head to enhance the convergence of SCB in groups $\mathbb{D}$ and $\mathbb{E}$, we are able to improve performance by 0.88% and 1.35% compared to group $\mathbb{C}$. Finally, group $\mathbb{F}$ incorporates the boundary consistency loss from group $\mathbb{E}$, and we can conclude that this loss can effectively enhance the consistency between the bordering regions of adjacent patches, up to 75.44% mIoU. Additionally, when combining the logits maps from the surrounding branch and the local branch, the performance is further improved to 75.59% mIoU. The improvement in results essentially belongs to a model ensembling method. This indicates that our surrounding branch has learned effective and complementary information to the local branch, further demonstrating the validity of our approach.

Other than analyzing the effectiveness of individual components within the module, we also demonstrate its generality across different methods, as shown in Fig. 5. As observed, on the general dataset such as Cityscapes, our method still lead to significant and consistent improvements upon various models. In addition, we select objects whose area is less than 900 pixels (0.0036% of the whole image) from Inria Aerial dataset to verify the effect on small objects. Adding SCB improved the mIoU from 57.12% to 57.73%, which shows SCB also improve the tiny objects.

### 4.4.2 Efficacy of Surrounding Context Integration Module

To explore how the attention mechanism and global interaction affect the performance, we conduct ablation studies on surrounding context integration module in Table 3. We first analyze the difference between the naive self-attention (Naive SA) and the window self-attention (W-MSA) in modeling the attention information of the surrounding patch. We apply six residual layers to replace attention module as the baseline, achieving 74.40% mIoU. Building on this, we further utilize Naive SA and W-MSA, which improve the mIoU by 0.46% and 0.32%, respectively. This indicates that the attention

mechanism can capture the correlation between different positions, promoting the flow of surrounding information within the feature map. As naive self-attention operates on every pixel of the feature map, it inherently incorporates surrounding interaction operation. We further analyze the impact of incorporating surrounding information into the W-MSA mechanism, and we utilize convolution and GAP operations to extract surrounding information from the windows. After adopting either convolution or GAP, we can get consistent improvement, indicating that introducing surrounding context is essential for W-MSA, which enables the current window to acquire information of other regions. In comparison to the convolution, using GAP to extract the surrounding representation of the window is more suitable for W-MSA, which can reach up to 75.44% mIoU.

### 4.4.3 Efficacy of Feature Fusion

We perform experiments to analyze the scheme of feature fusion of SCB and general segmentation module, as shown in Table 4. The fusion position including before the decoder head (early fusion) and after the decoder head (late fusion). The fusion method including ADD and CONCAT. We observe that the performance of late fusion is better than early fusion. We believe that late fusion is more flexible, shielding the complex processing of different models in the input part of the decoder head. Both ADD and CONCAT operations yield satisfactory outcomes, where the latter have a leading of 0.51%. We believe that the segmentation head captures a greater amount of information, allowing it to dynamically select the optimal feature for prediction. Hence, modifying the segmentation head of GSS can further enhance the performance of model, but it deviates from our starting point of simplicity and is outside the scope of our method.

### 4.4.4 Comparison of Surrounding Patch Size

We perform ablation studies on the surrounding patch size in Table 5. We observe that, as the surrounding size increases, the performance initially improves and then gradually decreases. This is attributed to the fact that only those close nearby area could provide valuable contextual guidance. Too far-away pixels in the entire images are not relevant to the local patch and may introduce additional noise. This confirms the motivation that contextual information around the local patch is beneficial.

### 4.4.5 Efficiency Study

We conduct experiments to examine the speed of different methods. Frames-per-second (FPS) and Memory are measured on a Tesla V100 GPU with a batch size of 1. The variation in FPS among different methods is primarily attributed to the inference framework, that is, whole inference and slide inference. Without considering the overlap, the latter theoretically requires $\lceil \frac{W}{w} \rceil \times \lceil \frac{H}{h} \rceil$ more operations than the former. Methods such as ISDNet primarily focus on model efficiency and achieve higher FPS, based on a shallow-deep architecture that directly employs the whole image for inference. Consequently, we also develop a fast version named SGNet (ISDNet-Style) by incorporating a modified SCB for a fair comparison. SGNet (ISDNet-Style) applies a single surrounding context integration module to the output feature map of the last stage of STDC to model global information. To maintain consistency with ISDNet and exclude speed differences caused by inference frameworks, we set our local patch size as same as ISDNet to simulate the whole inference. As shown in Table 6, the fast version not only surpasses the mIoU of ISDNet but also achieves a higher FPS of up to 25.59.

Table 6: Comparison of speed on DeepGlobe. We measure GPU memory using the command line tool "gpustat". "∗" represents results we reproduced in our setting. "-" indicates that there is no publicly available result or code.

| Method | mIoU | FPS | Memory(MB) |
|---|---|---|---|
| GLNet [6] | 71.60 | 0.17 | 1865 |
| CascadePSP [8] | 68.50 | 0.11 | 3236 |
| PPN [45] | 71.90 | 12.90 | 1193 |
| PointRend [23] | 71.78 | 6.25 | 1593 |
| MagNet [16] | 72.96 | 0.80 | 1559 |
| MagNet-Fast [16] | 71.85 | 3.40 | 1559 |
| FCtL [24] | 72.76 | 0.13 | 4332 |
| ISDNet [14] | 73.30 | 22.67* | 1948 |
| GPWFormer [19] | 75.80 | - | 2380 |
| DeepLabV3Plus [4] | 73.22 | 1.14 | 1279 |
| DeepLabV3Plus + *SGNet* | 75.44 | 0.66 | 2187 |
| *SGNet* (ISDNet-Style) | 74.28 | 25.59 | 2043 |

Table 7: Comparison of normal image and JPEG compressed image.

| | Normal | JPEG compression |
|---|---|---|
| DeepLabV3Plus + *SGNet* | 73.22 | 60.24 |
| | 75.44 (+2.22) | 64.16 (+3.92) |

#### 4.4.6 Robustness Study

We compressed the DeepGlobe dataset to 10% of its original image quality using JPEG compression and retrained SGNet and DeepLabV3Plus on it. As shown in Table 7, the results show that SGNet significantly outperforms DeepLabV3Plus by 3.92% (from 64.16% mIoU versus 60.24% mIoU), and this improvement is almost twice that of normal images (from 73.22% mIoU to 75.44% mIoU). This indicates that our method is relatively insensitive to noise compared to baseline models and can use surrounding information to infer damaged pixel information within the object. It also demonstrates that our method is particularly robust in scenarios involving image degradation.

#### 4.4.7 Fragmentation Phenomenon Study

To show our advantage to alleviate the fragmentation phenomenon, we compare our module against GSS in Fig. 6. We simulate all predicted patches (non-overlapping) generated during slide inference process with white lines. It is evident that GSS result in a steep change in the boundary between adjacent patches, and the predicted results of each patch are relatively independent, lacking coherence. Our method is capable of modeling the correlation between patches, leading to smoother prediction results for adjacent patches.

## 5 Conclusion

In this paper, we propose the Surrounding Guided Segmentation Framework (SGNet) to address the scalability and architectural issues in existing UIS methods. SGNet leverages surrounding context to guide local patch segmentation and can be incorporated into any general segmentation model. Our method consistently improves performance across five datasets and demonstrates greater adaptability and applicability than existing methods in real-world scenarios. This work not only contributes a novel solution to the UIS domain but also emphasizes the potential of integrating general segmentation techniques to advance the field. We hope that SGNet inspires further exploration in ultra image segmentation, fostering innovations that enhance model performance and scalability.

**Limitations.** Noise and artifacts in ultra-high-resolution images can hinder segmentation accuracy, necessitating further research to address these challenges.

**Social Impact.** The proposed method has the potential to advance various fields, including medical image analysis and remote sensing image processing.

**Acknowledgment.** This work was supported in part by the National Key Research and Development Program of China under grant 2023YFC2705700, in part by the National Natural Science Foundation of China under grant 62372341, and in part by the Fundamental Research Funds for the Central Universities under grant 2042024kf0040.

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

# A    Existing Architecture Analysis

In Figure 7, we show the detail architectures of existing ultra image segmentation methods. Whole inference 7(a) and slide inference 7(b) are the two most fundamental ultra image segmentation architectures. While whole inference can be applied directly to general segmentation models [27, 4, 38, 46, 12, 49], it utilizes the down-sampled image as input, which results in a loss of essential information. The latter divides the image into multiple patches by means of sliding window, and predicts the patches one by one, aggregating the results together. However, due to the isolated prediction of patches and the lack of global information, it is prone to cause **information bottleneck** and **fragmentation phenomenon**.

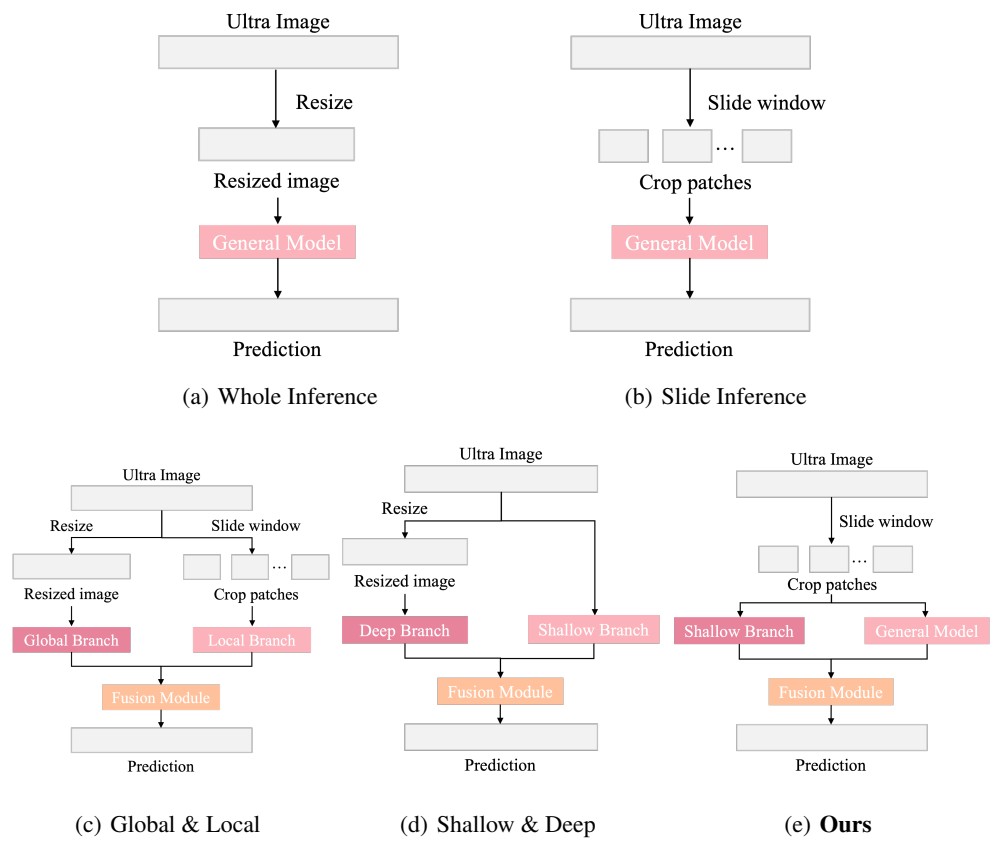

Figure 7: Comparison of existing architectures of ultra image segmentation.

To address the issues above and leverage the benefits of both whole inference and slice inference, some studies [6, 45, 24, 16, 19] have introduced the Global & Local architecture 7(c). This architecture involves one branch that takes in the down-sampled ultra image as the global cue, while the other branch extracts local information from sliced patches. However, these tasks usually require sequential processing of each patch and complex integration strategies to combine global information, resulting in poor scalability of the architecture. Alternatively, some study[14, 20] have introduced the Shallow & Deep architecture 7(d), which inputs the original image and the resized image into different branches for processing. Despite its fast speed, it is essentially a trade-off in dataset size and computing resources, and cannot be used for real ultra image segmentation scenarios.

The architectures depicted in Figure 7(c) and Figure 7(d) both use the original image after resize in an input branch, which may limit the capability of the architecture to process ultra images. On the one hand, when the image is extremely large, the resized image will inevitably lose a lot of detail information to fit the GPU memory. On the other hand, the resized ultra image cannot guarantee to provide specific global information that is helpful for local patch segmentation. To address these issues, we designed a novel and effective ultra image segmentation architecture 7(e) based on slide

window. Our architecture can handle ultra images of any scale and provide surrounding information that aids in local patch segmentation. Furthermore, our architecture is highly flexible, not restricted by computing resources, and can be seamlessly integrated to any encoder-decoder based segmentation model.

## B    Dataset Details

To comprehensively evaluate our method, we conduct experiments on five public ultra image datasets involving general, medical and remote sensing scenarios:

**Cityscapes [10].** The Cityscapes dataset is a popular street scene dataset for generic semantic segmentation. It contains 3475 images with the resolution of 1024×2048 and a total of 19 categories. We use 2975 images for training and 500 images for testing.

**Gleason [21].** The Gleason dataset is a high resolution medical image dataset with the resolution of 5120×5120. It contains 244 H&E-stained histopathology images for automatic Gleason grading of prostate cancer. Since we do not have data partition details from [16], we randomly split dataset into training and testing set with 195 and 49 images.

**DeepGlobe [11].** DeepGlobe is a satellite image dataset that contains 803 ultra-high resolution images (2448×2448 pixels). It contains 7 categories, of which the class named "unknown" is excluded. Following [6], we divide the training, validation, and testing sets into 454, 207, and 142 images, respectively.

**Inria Aerial [30].** The Inria Aerial dataset comprises 180 images, each with a resolution of 5000×5000 pixels. Following [6], we divide the training, validation, and testing sets into 126, 27, and 27 images, respectively.

**Five-Billion-Pixels [40].** The FBP dataset includes 150 high-resolution images, each with a size of 7200×6800 pixels and labeled with 24 categories. We use the same test set as in [40], and randomly divide the remaining images into training set and validation set, including 90 and 30 images.

## C    Comparative Analysis of SCB Branch

In order to demonstrate the proposed SCB Branch's efficacy, we used sixteen conventional convolution layers followed by four transformer blocks to form a trivial replacement branch for extracting surrounding image feature. It serving as a functionally analogous replacement for the proposed SCB branch. As shown in Table 8, we added this trivial replacement branch to all general segmentation models in Table 1 and conducted experiments on the DeepGlobe dataset using the same settings. The results show that our proposed SCB branch significantly outperforms this branch, proving the efficacy of our proposed component.

Table 8: Comparison of SCB Branch with it trivial replacement.

|  | Original | + Trivial Branch | + SGNet |
|---|---|---|---|
| FCN | 72.38 | 72.56 (+0.28) | 75.28 (+2.90) |
| DeepLabV3Plus | 73.22 | 73.77 (+0.55) | 75.44 (+2.22) |
| HRNet | 72.87 | 73.24 (+0.37) | 75.25 (+2.38) |
| SegFormer | 72.96 | 73.56 (+0.60) | 74.65 (+1.69) |
| STDC | 72.59 | 72.88 (+0.29) | 74.51 (+1.92) |

## D    Large Scale Human Subject Segmentation Study

We conducted experiments on the well-known CelebAMask-HQ dataset to further verify the effectiveness of our method in large scale human subject segmentation. Due to the lack of extremely high-resolution human datasets, we simulate ultra-high resolution by resizing the images from the CelebAMask-HQ dataset from 1024 to 2448 pixels. We compared our SGNet with DeepLabV3Plus, which is a highly popular and widely used image segmentation model across various domains. The performance of our method significantly outperformed DeepLabV3Plus by 1.61% (from 62.93% mIoU to 64.54% mIoU). We provide more examples of qualitative results in Figure 8.

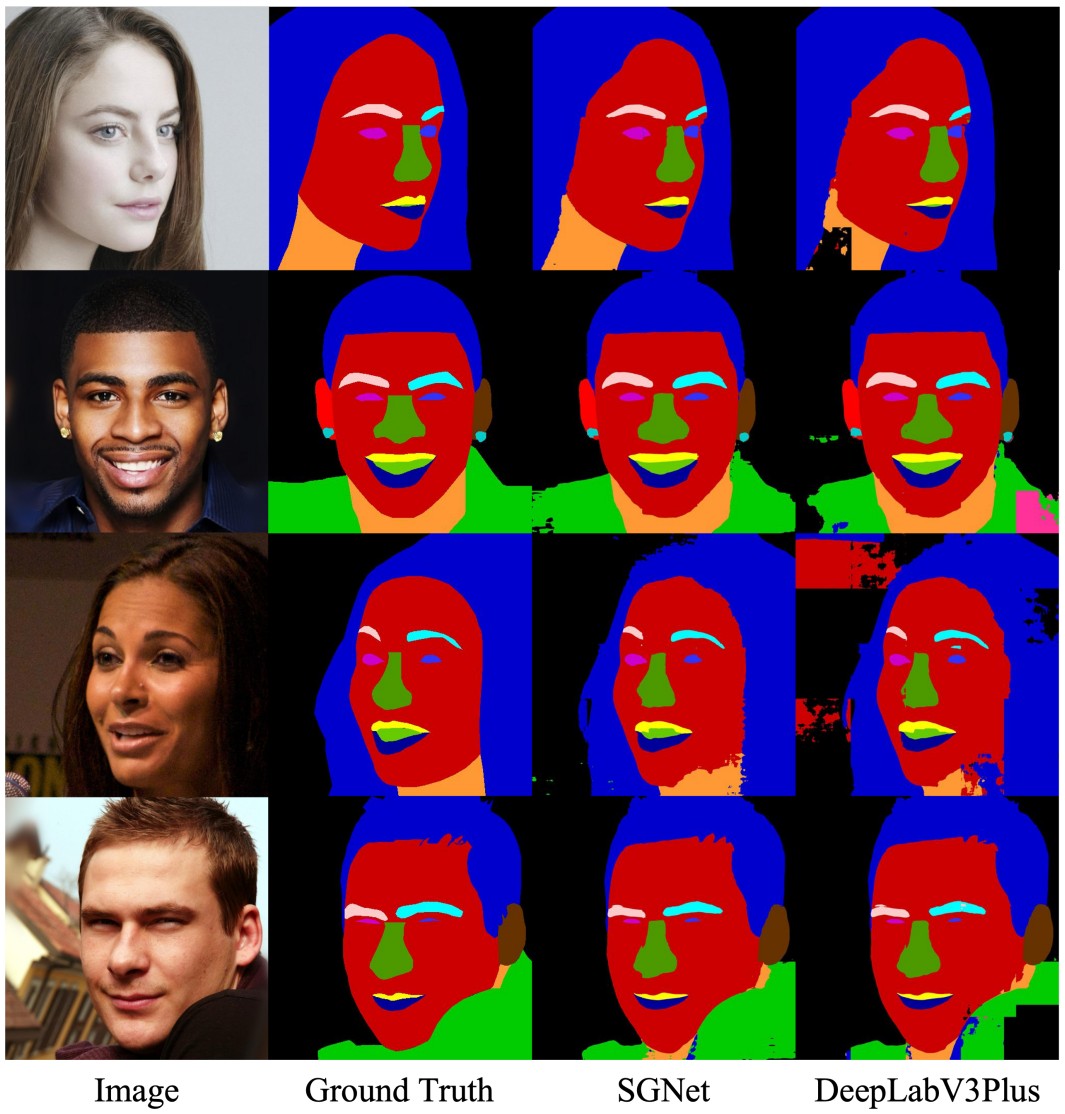

| Image | Ground Truth | SGNet | DeepLabV3Plus |

Figure 8: The visualization of different methods in CelebAMask-HQ. We resized the images from 1024 to 2448 pixels to simulate ultra-high resolution and used sliding window of size 512 without overlap for inference.

# E More Qualitative Result

In Figure 9, we provide more examples of qualitative results between our method and existing ultra image segmentation methods like FCtL [24] and ISDNet [14]. These results indicate that our method is capable of achieving satisfactory quality on various challenging datasets [11, 30]. Figure 10 and Figure 11 illustrate the effectiveness of SCB by comparing it to general segmentation models such as DeepLabV3Plus and HRNet on Cityscapes [10] and Gleason [41].

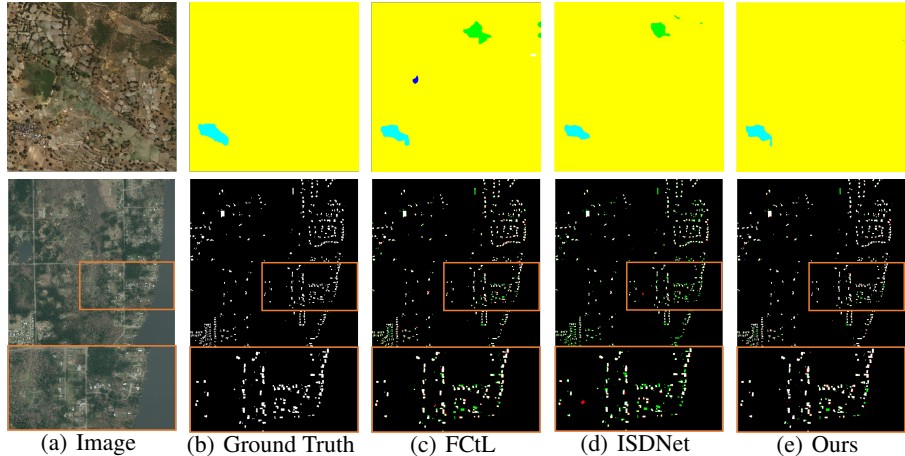

|        |                  |          |           |          |
|--------|------------------|----------|-----------|----------|
| (a) Image | (b) Ground Truth | (c) FCtL | (d) ISDNet | (e) Ours |

Figure 9: The qualitative results of different methods in *DeepGlobe* (top row, 2448×2448) and *Inria Aerial* (bottom row, 5000×5000).

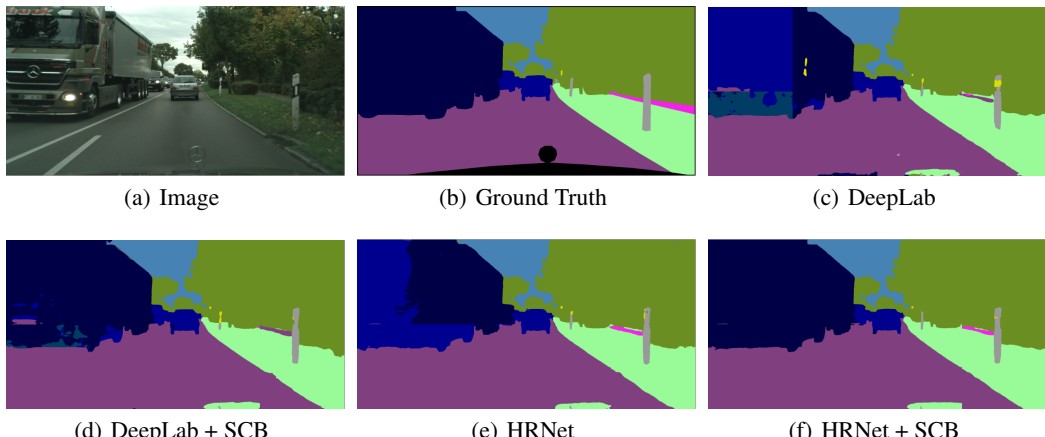

|          |                  |            |
|----------|------------------|------------|
| (a) Image | (b) Ground Truth | (c) DeepLab |
| (d) DeepLab + SCB | (e) HRNet | (f) HRNet + SCB |

Figure 10: The comparison of adding SCB on different models in *Cityscapes* (2048×1024).

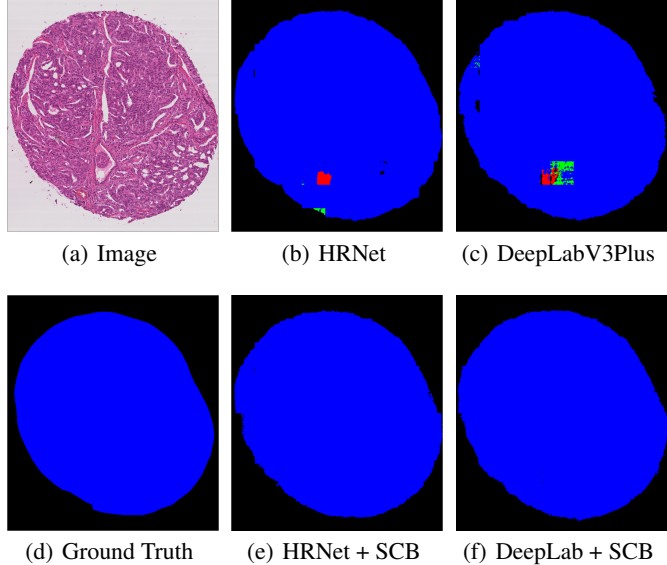

|          |            |                  |
|----------|------------|------------------|
| (a) Image | (b) HRNet | (c) DeepLabV3Plus |
| (d) Ground Truth | (e) HRNet + SCB | (f) DeepLab + SCB |

Figure 11: The comparison of adding SCB on different models in *Gleason* (5120×5120).

