# OpenReview forum: "Toward Real Ultra Image Segmentation: Leveraging Surrounding Context to Cultivate General Segmentation Model"
_NeurIPS.cc/2024/Conference — NeurIPS 2024 poster_

### Official Review · Reviewer_xcUS · 2024-07-01

**Soundness:** 4
**Presentation:** 4
**Contribution:** 4
**Rating:** 7
**Confidence:** 4

**Summary:**

The paper discusses the challenges of ultra image segmentation and proposes a Surrounding Guided Segmentation framework (SGNet) to address these challenges by leveraging surrounding context information.

**Strengths:**

1) Novelty: The paper identifies two specific issues to ultra image segmentation, namely generalization and architectural issues, and proposes a solution to solve both. The proposed SGNet is quite novel and the biggest plus is that it can be used alone with any segmentation model to improve the quality of the results. This can be beneficial in a variety of image segmentation applications.
2) Evaluation: The proposed method has been validated on a variety of datasets, which helps build confidence in the method. The method also consistently seems to outperform previous methods.
3) The contribution resulting from each individual part of the module has been clearly evaluated, which explains the efficacy of each.
4) The method can handle images of any scale.

Overall it seems to be a solid paper in my opinion with sufficient novelty.

**Weaknesses:**

1) For some reason, there seems to be no evaluation done on human subjects specifically. I would have liked to see some segmentation results on human subjects, especially around fingers, ear etc.
2) The discussion on limitations is extremely limited. Please read my comment in that section.

**Questions:**

1) In equation 7 for the loss function, the paper uses 3 parameters, where we could only do with 2 after normalization.
2) It is typical to represent L1 loss with a subscript for clarity, so I would add it in equation 6.
3) The paper uses the word performance in several instances to discuss quality. It is an important distinction, as the usage of performance should be limited to quantify speed, and quality should be used for discuss the improvements in segmentation mask quality.
4) Table 6 discusses the memory footprint of the method, but it is quite unclear how this is measured or what it represents. Is it peak memory usage? Average memory usage during inference?
5) In the efficiency study, it is unclear what device is the method evaluated on.
6) " The method can handle images of any scale.", this is only partially true as you are still going to be restricted by your compute resources. This needs to be clarified.

**Limitations:**

The discussion on limitations is extremely limited. Can this method be used for matting for instance? Or very thin peninsular regions? Will it improve or reduce the quality of segmentation if the images have jpeg artifacts? Can it be used on difficult objects like jewelry? These questions could be discussed in the limitations section.

---

> ### Author Rebuttal · Authors · 2024-08-07
>
> > W1: I would like to see results on human subjects.
>
> We aim to address the task of ultra image segmentation and common academic datasets are typically in remote sensing or medical imaging (L183-186).
> To further verify the effectiveness of our method in large scale human subject segmentation, we conducted experiments on the well-known CelebAMask-HQ dataset[1].
>
> Due to the lack of extremely high-resolution human datasets, we simulate ultra-high resolution by resizing the images from the CelebAMask-HQ dataset from 1024 to 2448 pixels.
> We compared our SGNet with DeepLabV3Plus, which is a highly popular and widely used image segmentation model across various domains.
>
> The performance of our method significantly outperformed DeepLabV3Plus by 1.61 (from 62.93 mIoU to 64.54 mIoU).
> Please refer to our attached pdf for visualization.
>
> [1]: Lee, Cheng-Han, et al. "Maskgan: Towards diverse and interactive facial image manipulation." CVPR. 2020.
>
> > Q1: In equation 7 for the loss function, the paper uses 3 parameters, where we could only do with 2 after normalization.
>
> Thank you for your suggestion. Yes, it can be reformulated with two hyper-parameters. We will revise this function in the final version.
>
> > Q2: It is typical to represent L1 loss with a subscript for clarity, so I would add it in equation 6.
>
> Thank you for pointing out this issue. We have revised Equation 6 to the following form: ${L} _ {consistency}=||{P}^{'} _ {global}-{P} _ {local}{||} _ {1}$.
>
> > Q3: The paper uses the word performance in several instances to discuss quality.
>
> We apologize for the inaccurate use of the terms 'performance' and 'quality'. We have revised the relevant sections and will update the final version.
>
> > Q4: Explanation of 'memory' in Table 6，peak or average?
>
> Apologies for the missing explanation. 'Memory' refers to the maximum GPU memory consumption during model inference. We measure GPU memory using the command line tool 'gpustat', following previous works such as ISDNet.
>
> > Q5: what device is the method evaluated on.
>
> We evaluated on a Tesla V100 GPU with a batch size of 1.
>
> > Q6: The method can handle images of any scale.", this is only partially true as you are still going to be restricted by your compute resources. This needs to be clarified.
>
> Not agreed.
>
> One of our highlights is the ability to handle images of any scale. Our approach only uses sliding window mode for predictions, making it independent of computational resource limitations. Our method uses the same amount of GPU memory for handling 2448x2448 and 6800x7200 images.
>
> > L1: Could this method be used for image matting, very thin peninsular regions, or difficult objects like jewelry? Will it improve segmentation quality with JPEG artifacts？ Need to be discussed in the limitation section.
>
> (1) Our method is primarily designed for ultra image segmentation, with its core approach utilizing surrounding information to guide local patch segmentation in the sliding window mode.
>
> (2) For image matting tasks, accurate segmentation of instance boundaries is crucial. Our method, leveraging surrounding information, effectively enhances boundary segmentation quality, as demonstrated in Figure 6 of the original manuscript. It is also applicable to matting for ultra image and addresses the issue of inconsistent foreground instances between adjacent image patches.
> Moreover, our method can be seamlessly integrated into most image matting models.
>
> (3) Our method is suitable for handling narrow peninsula regions, which also appear in the DeepGlobe and FBP datasets.
>
> (4) We compressed the DeepGlobe dataset to 10\% of its original image quality using JPEG compression and retrained SGNet and DeepLabV3Plus on it.
> The results show that SGNet significantly outperforms DeepLabV3Plus by 3.92 (from 64.16 mIoU versus 60.24 mIoU), and this improvement is almost twice that of normal images (from 73.22 mIoU to 75.44 mIoU).
> This indicates that our method is relatively insensitive to noise compared to baseline models and can use surrounding information to infer damaged pixel information within the object. It also demonstrates that our method is particularly effective in scenarios involving JPEG compression.
>
> (5) Our work is independent of the difficulty of the objects and is more focused on their performance in ultra images. Thus we may not be different from other models.

---

### Official Review · Reviewer_HefE · 2024-07-08

**Soundness:** 3
**Presentation:** 3
**Contribution:** 3
**Rating:** 3
**Confidence:** 5

**Summary:**

The paper introduces SGNet to address the limitations of existing UIS methods, i.e., generalization issues and architectural constraints. SGNet leverages a larger context around image patches to refine segmentation results, making it compatible with general segmentation models and enhancing their performance on high-resolution images. The framework is validated on five diverse datasets, demonstrating significant improvements in segmentation accuracy and efficiency.

**Strengths:**

(1) This method proposes SGNet to address two intrinsic problems in UIS.
(2) The method is evaluated on a wide range of datasets, including Cityscapes, DeepGlobe, Inria Aerial, Five-Billion-Pixels, and Gleason, showcasing its effectiveness across various scenarios, including medical and remote sensing applications.

**Weaknesses:**

(1) The challenges in Contribution 1 have been proposed in many segmentation tasks. Therefore, C1 is weak.
(2) The contribution of the SGNet is incremental, including the consistency loss and the local and global branches.
(3) If the author wants to verify the SGNet is plug-and-play, the author should compare it with other plug-and-play methods rather than compare it with other task-specific strategies.
(4) The computational cost is expensive.

**Questions:**

see weakness

**Limitations:**

see weakness

---

> ### Author Rebuttal · Authors · 2024-08-07
>
> > W1: The challenges in Contribution 1 have been proposed in many segmentation tasks. Therefore, C1 is weak.
>
> The contribution 1 is: "We excavate two essential but largely overlooked issues in UIS, which hold great value for the community. In addressing these challenges, we are the first to tackle the ultra image segmentation task from the general segmentation model perspective" (L65-67).
> The two issues represent the **Generalization issue** and **Architectural issue**.
>
> Previous works have all been specialized UIS methods (Generalization issue). We are the first to address the task of ultra image segmentation from the perspective of general segmentation model.
> The architectures of previous works are not suitable for handling images of extremely large scales in real-world scenarios (Architectural issue).
>
> Therefore, the two issues presented in Contribution 1 are unique to the task of ultra image segmentation.
> As far as we know, we are the first to propose the above viewpoint. Please provide any reference if we miss.
>
> > W2: The contribution of the SGNet is incremental, including the consistency loss and the local and global branches.
>
> Not agreed.
>
> The contribution of SGNet is that we are the first to address ultra image segmentation from the perspective of general segmentation models, capable of handling images of extremely large scale.
> It is plug-and-play, can be integrated into any general segmentation model (we tested five different GSS models), and has consistently improved performance across multiple datasets.
>
> This has been epoched by the other reviewers that our method is "an interesting approach to the problem" (Reviewer-MJRN) and "quite novel" (Reviewer-xcUS). They noted that SGNet "is simple and effective" (Reviewer-1ux8), and "the biggest plus" (Reviewer-xcUS) is that it "can be integrated with various general segmentation models" (Reviewer-MJRN).
>
> > W3: If the author wants to verify the SGNet is plug-and-play, the author should compare it with other plug-and-play methods rather than compare it with other task-specific strategies.
>
> Our proposed SGNet can be integrated into any segmentation model, and we have conducted experiments on segmentation models with five different models, including CNN, Transformer, and lightweight models architectures (L202-204), to demonstrate that our method is "plug-and-play".
>
> Our paper targets the ultra image segmentation task.
> Thus, we proposed SGNet to migrate general segmentation models to this task due to its better scale-up ability.
> Therefore, it is appropriate for us to compare our method with other UIS task-specific methods.
>
> We are the first to offer a "plug-and-play" method for ultra image segmentation. As far as we know, there are no "other plug-and-play methods" for this task.
>
> Please list any reference if you found.
>
> > W4: The computational cost is expensive.
>
> Our main speed bottleneck lies in the sliding window architecture.
> Although using the entire image as input processes the image only once,
> it can only handle images of limited size.
> In contrast, our method processes the same image multiple times but it is the only way to handle images of extremely large scales (L36-43).
> We also develop a fast version for a fair comparison with those methods that emphasize speed, and we surpass the comparison methods in both speed and performance (L292-299).
>
> Furthermore, we believe that the UIS community prefer prioritizes usability and quality in real-world scenarios over speed requirements like real-time.
> In practical applications, there are numerous engineering optimization techniques, such as model quantization and multi-GPU parallelism, that can enhance speed.
>
> At the same time, UIS methods typically have more complex architectures and present deployment challenges (L29-32).
> In contrast, our method can be seamlessly integrated into general segmentation models, making deployment faster and easier.

---

> > ### Comment · Reviewer_HefE · 2024-08-11
> > **Official Comment by Reviewer HefE**
> >
> > I appreciate the author's feedback. I am inclined to maintain my score because my major concerns, including incremental contributions and insufficient experiments, haven't been addressed.

---

> > > ### Author Response · Authors · 2024-08-12
> > >
> > > Our listed contributions do not include anything related to "the consistency loss and the local and global branches".
> > > Instead, we emphasize the framework that leverages surrounding context information to guide a general segmentation model in segmenting local patches. It also can be integrated into any general segmentation model.
> > > Furthermore, we are the first to address the UIS task from the perspective of a general segmentation model, which is different from all previous works.
> > >
> > > We've already explained W3 (insufficient experiments) in our rebuttal, clarifying that we are the first to propose a "plug-and-play" method for ultra image segmentation, and there are no other "plug-and-play methods" for this task. Please list any references if we missed and specify what "insufficient experiments" includes.

---

### Official Review · Reviewer_MJRN · 2024-07-12

**Soundness:** 3
**Presentation:** 2
**Contribution:** 3
**Rating:** 5
**Confidence:** 5

**Summary:**

To overcome the challenges in generalization and compatibility with real-world ultra images, SGNet revisits a classic sliding inference approach and incorporates a surrounding context module to refine local patch segmentation. SGNet is compatible with various segmentation models and achieves significant performance improvements.

**Strengths:**

* The authors address ultra image segmentation from the perspective of general segmentation models. This is an interesting approach to the problem.
* The proposed SGNet leverages surrounding context information to enhance the segmentation results obtained from analyzing image patches.
* This framework can be integrated with various general segmentation models and achieves competitive performance across different general segmentation models on five publicly available datasets.

**Weaknesses:**

1. The authors emphasize in lines 29 to 35 that general segmentation models possess superior generalization capabilities. However, from the results presented in Table 1, it can be observed that even general segmentation algorithms exhibit significant performance fluctuations across different datasets. This is particularly evident on the Cityscapes dataset.
2. Regarding the experiments detailed in Table 1, several general segmentation methods outperform existing specialized ultra image segmentation approaches on DeepGlobe, FBP, and Aerial Inria datasets without introducing the proposed SGNet.
    1. The authors attribute their performance superiority to the generalizability or architecture. However, they overlook the impact of different inference modes. Based on the description provided in the paper, all general segmentation methods employ sliding inference. However, this is not consistent across existing methods. This discrepancy precludes a direct conclusion that "general segmentation algorithms have better generalization."
    2. Given the diversity of inference modes, the authors should supplement Table 1 with specific inference settings for each method to identify more critical factors.
3. The authors mention in line 489 that due to the inaccessibility of the data partition details from [1], they conducted their own dataset division. Yet, in Table 1, only ISDNet [4] presents results for Gleason. This is logically inconsistent. Did the authors retrain ISDNet using your own data split?
4. The validation of the proposed component's efficacy is not rigorous. Since the component is pluggable, the performance evaluation should compare two variants: 1) removing the proposed component directly as shown in Table 1, and 2) replacing it with a trivial operation of similar function, such as some simple convolutional layers with the same number of parameters, or components with analogous functions from existing methods. The latter was overlooked by the authors.
5. The design of the surrounding context integration module appears strikingly similar to that in [2]. The authors should provide a detailed comparison.
6. There is a citation error for the Five-Billion-Pixels dataset. It should be referenced as [3].
7. For the experiments in Table 2:
    1. The order is illogical. It would be natural to integrate the alignment operation into the model before introducing SCI.
    2. The boundary consistency loss yields a limited performance gain.
8. What is the specific architecture of the first model variant listed in Table 3?

**Questions:**

See Weaknesses.

---

> ### Author Response · Authors · 2024-08-05
> **Request for clarification for W5 "reference [2]"**
>
> Dear Reviewer,
>
> We are currently in the process of drafting our rebuttal response and would greatly appreciate your clarification on a point mentioned in Weakness 5.
>
> Could you please provide the specific title of the reference [2] so that we can address this point more accurately in our response.
>
> Thank you very much for your assistance!
>
> Sincerely,
> Authors

---

> ### Author Rebuttal · Authors · 2024-08-07
>
> > W1: The authors emphasize that general segmentation models possess superior generalization capabilities, but they still exhibit significant performance fluctuations.
>
> Sorry for misleading.
>
> We use the term "generalization ability" to represent a model's capability to scale up to larger resolutions, rather than performance fluctuations across datasets.
> Our intention was to highlight that ultra image segmentation (UIS) methods struggle to scale up to larger image sizes, rather than the variance of the same method across different datasets.
> As image resolution increases (e.g., from 2448x2448 to 6800x7200), the performance of UIS methods significantly degrades (GLNet, a typical UIS method, drops from 71.60 to 42.05, due to processing the entire image information).
> In contrast, our method is designed to handle extremely high resolutions effectively and is less affected by image size (e.g., from 75.44 at 2448x2448 resolution to 63.18 at 6800x7200 resolution), making it more suitable for extremely large image.
> In the revised version, we will use more precise terms to avoid ambiguity.
>
> > W2: The inference mode of general segmentation methods is not consistent across existing UIS methods and affects the performance.
>
> We have supplemented Table 1 of the original manuscript with specific inference modes for each method (see attached pdf) and extracted UIS results on the DeepGlobe dataset as follows.
>
> Most UIS methods in Table 1 use sliding window inference, so there is no issue with inconsistency in inference modes.
> From the comparison of the nine UIS methods, the sliding window mode does not have an advantage over whole inference.
> In addition, we reproduced the results of ISDNet by changing from whole inference to sliding window inference, and its performance even dropped.
> Moreover, the sliding window mode is more suitable in real applications, as it can scale up to extremely large images.
>
> | Method     | Mode        | DeepGlobe |
> | ---------- | ----------- | --------- |
> | ISDNet     | Whole       | 73.30     |
> | WSDNet     | Whole       | 74.10     |
> | ElegantSeg | Whole       | 74.32     |
> | ISDNet\*   | Slide       | 68.82     |
> | FCtL       | Slide       | 72.76     |
> | GLNet      | Slide+Whole | 71.60     |
> | PPN        | Slide+Whole | 71.90     |
> | MagNet     | Slide+Whole | 72.96     |
> | GPWFormer  | Slide+Whole | 75.80     |
>
> Table 1: Comparison with UIS methods. "\*" represents results we reproduced in the sliding window mode.
>
> > W3: Did the authors retrain ISDNet using your own data split.
>
> Yes, we retrained ISDNet using the same data split as ours.
>
> We apologize for the omission of specific experimental settings. ISDNet is the state-of-the-art model among all the **_open-sourced_** methods.
> We followed the original settings of ISDNet and retrained it on the Gleason dataset using our own data split.
>
> > W4: Extra experiments using trivial operations.
>
> Thank you for pointing out this issue. We have supplemented this comparative experiment to demonstrate the proposed component's efficacy in the below table.
> Specifically, we used sixteen conventional convolution layers followed by four transformer blocks to form a trivial replacement branch for extracting surrounding image feature, serving as a functionally analogous replacement for the proposed SCB branch.
>
> We added this trivial replacement branch to all general segmentation models in Table 1 of the original manusript and conducted experiments on the DeepGlobe dataset using the same settings.
> The results show that our proposed SCB branch significantly outperforms this branch, proving the efficacy of our proposed component.
>
> |               | Original | + Trivial replacement branch |    + SGNet    |
> | :-----------: | :------: | :--------------------------: | :-----------: |
> |      FCN      |  72.38   |        72.56 (+0.28)         | 75.28 (+2.90) |
> | DeepLabV3Plus |  73.22   |        73.77 (+0.55)         | 75.44 (+2.22) |
> |     HRNet     |  72.87   |        73.24 (+0.37)         | 75.25 (+2.38) |
> |   SegFormer   |  72.96   |        73.56 (+0.60)         | 74.65 (+1.69) |
> |     STDC      |  72.59   |        72.88 (+0.29)         | 74.51 (+1.92) |
>
> > W5: Provide a detailed comparison with [2].
>
> Sorry, we don’t know which paper [2] refers to. Could you provide the specific title? We are happy to provide more detailed comparisons.
>
> > W6: Citation error for the Five-Billion-Pixels dataset.
>
> Thank you for pointing out this error.
> We have updated the reference to "Enabling country-scale land cover mapping with meter-resolution satellite imagery. ISPRS Journal of Photogrammetry and Remote Sensing, 196:178–196, 2023."
>
> > W7: Ablation order and the performance of the boundary consistency loss.
>
> (1) Thank you for pointing out this issue. We conducted an experiment using the backbone with the alignment operation and achieved a 73.88 mIoU.
> We will add this comparison in the final version.
>
> (2) This is an auxiliary loss, which we did not list as a contribution. We initially hoped would encourage consistency in predictions across different branches for the same region. In practice, its constraint effect under supervised conditions is relatively limited.
>
> > W8: The architecture of the first model in Table 3.
>
> Thanks for remindering.
> The first model in Table 3 uses six residual layers (Conv-BN-ReLU) to replace the attention module. It takes feature maps divided into non-overlapping regions as input, allowing for simple and direct integration of surrounding features.
> In contrast, using attention mechanisms and global interaction allows us to selectively learn contextual information that benefits local patch segmentation.

---

> ### Comment · Reviewer_MJRN · 2024-08-12
> **Missing References in My Review**
>
> REF:
>
> * [1]: Progressive semantic segmentation, CVPR 2021
> * [2]: RegionViT: Regional-to-Local Attention for Vision Transformers, ICLR 2022
> * [3]: Enabling Country-Scale Land Cover Mapping with Meter-Resolution Satellite Imagery, ISPRS Journal of Photogrammetry and Remote Sensing 2023
> * [4]: Isdnet: Integrating shallow and deep networks for efficient ultra-high resolution segmentation, CVPR 2022

---

> > ### Author Response · Authors · 2024-08-12
> >
> > > W5: Provide a detailed comparison with [2].
> >
> > First, our surrounding patch is designed to assist the local patch. It is centered around the local patch and its area is twice as large. The surrounding patch dynamically changes with the local patch, providing sufficient contextual information and preventing fragmentation.
> > In contrast, the regions in RegionViT [2] are predefined based on the size of the image.
> > Dynamically adjusting the surrounding patch provides better information, while the fixed division in RegionViT can lead to even more fragmented prediction results. This is especially problematic for ultra image segmentation tasks, such as remote sensing images, where regional features can vary significantly.
> >
> > Second, the RegionViT integrates local tokens using self-attention with a single corresponding regional token. In contrast, we use the ROI Align operation to associate and merge information, which is designed for pixel-level understanding tasks, allowing precise alignment of information between the surrounding patch and the local patch.
> >
> > Third, our approach employs the global average pooling and broadcasting operations for exchanging information. It is non-parametric and simpler compared to RegionViT, which introduces additional attention parameters for local self-attention.
> >
> > Technically, we have not emphasized that the design of specific modules is our contribution, although the utilization of context information is similar across methods. The core contribution of our paper is addressing ultra image segmentation from the perspective of general segmentation models and proposing a plug-and-play approach that leverages surrounding information to guide a general segmentation model in segmenting local patches.

---

> ### Comment · Reviewer_MJRN · 2024-08-13
>
> Thank you to the authors for responding to my questions and basically put my concerns to rest. Considering the good results and reference value of this context-enhanced design, I am willing to raise the score.
>
> In fact, the existing review comments reflect the deficiencies that exist in the current version of the paper.
> Regardless of eventual acceptance into the conference, I hope that the authors will improve and add these discussed contents in the revised version.

---

> > ### Author Response · Authors · 2024-08-13
> >
> > Dear Reviewer MJRN:
> >
> > We greatly appreciate your suggestions!
> >
> > Your comments are highly valuable and have provided excellent insights for our paper.
> > We will definitely incorporate all the discussed suggestions in the revised version, which will further highlight the strengths of our work.
> >
> > Sincerely, Authors

---

### Official Review · Reviewer_1ux8 · 2024-07-12

**Soundness:** 3
**Presentation:** 3
**Contribution:** 3
**Rating:** 7
**Confidence:** 4

**Summary:**

This paper focus on the generalization and architectural issues of the ultra image segmentation methods, and proposes SGNet which consists of two branches for processing surrounding patch and local patch, respectively. The motivation is to leverage the surrounding context for refining the segmentation results of local patches. The results show a significant improvement over previous methods. Overall, the paper is well-written with good results.

**Strengths:**

(1) The paper is well-written and the figures and tables are well-organized and clear.

(2) The analysis of problems in the previous work is clear and logical, which is easy to follow.

(3) The proposed method is simple and effective and shows advantages on various datasets with various general segmentation models.

**Weaknesses:**

(1) The description of the surrounding branch is not clear enough (see question (1)).

(2) The evaluation of speed was insufficient (only included one of the lightest versions, i.e., ISDNet-Style). More adequate results could help to better understand the overhead of the method.

**Questions:**

(1) What kind of backbone network is used in surrounding branch? Is it the same as that in local branch? Does the surrounding branch resize its input?

(2) A sliding window without overlap is used for inference, does this (with or without overlap) make a difference to the gain of the method?

(3) According to Figure 1, the surrounding branch’s output is discarded during inference. Is it possible to further improve the performance by combining it with the result of the local branch?

(4) There is a typo at last of the caption of Figure 1.

**Limitations:**

The limitations and social Impact have been discussed.

---

> ### Author Rebuttal · Authors · 2024-08-07
>
> > W2: The evaluation of speed was insufficient (only included one of the lightest versions, i.e., ISDNet-Style). More adequate results could help to better understand the overhead of the method.
>
> We have given the full-version (SGNet) speed comparison in Table 6 of the original manuscript (L281-299). It is attached to the DeeplabV3Plus model as backbone.
>
> Some works, such as ISDNet, directly take the entire image or a downsampled version for prediction to achieve faster speed, but face the architectural issue (L36-41). When the image is excessively large (e.g., 7200x6800), the former (entire image input) is infeasible due to limited GPU memory, while the latter (downsampled image) suffers significant information loss during compression, making them unsuitable for ultra image segmentation in real-world scenarios.
>
> > Q1: What kind of backbone network is used in surrounding branch? Is it the same as that in local branch? Does the surrounding branch resize its input?
>
> (1) The surrounding branch and local branch use different backbones.
> We employ a lightweight backbone for the surrounding branch, i.e., the first four layers of STDC, in order to boost processing speed (L120-121, L190-191).
>
> (2) The surrounding branch does not resize the input, which helps to better enhance spatial details.
> It always maintains the original resolution, which is different from previous works that use downsampled images and suffer information loss when scaled up to extremely large resolution.
>
> > Q2: A sliding window without overlap is used for inference, does this (with or without overlap) make a difference to the gain of the method?
>
> All the methods reported in Table 1 of the original manuscript are sliding window without overlap, ensuring a fair comparison (L189-190).
>
> A sliding window with overlap is essentially a test time augmentation method, where multiple predictions on overlapping regions are averaged to enhance the model's performance.
> By using overlapping regions that cover half of the input patch for predictions, our method can further improve the performance by 0.24 (from 75.44 mIoU to 75.68 mIoU).
>
> > Q3: According to Figure 1, the surrounding branch’s output is discarded during inference. Is it possible to further improve the performance by combining it with the result of the local branch?
>
> Yes, it will.
>
> When combining the logits maps from the surrounding branch and the local branch, the performance is further improved by 0.15 (from 75.44 mIoU to 75.59 mIoU) on the DeepGlobe dataset.
> The improvement in results essentially belongs to a model ensembling method.
> This indicates that our surrounding branch has learned effective and complementary information to the local branch, further demonstrating the validity of our approach.
>
> > Q4: There is a typo at last of the caption of Figure 1.
>
> Thank you for pointing out this issue and we have fixed it.

---

> > ### Comment · Reviewer_1ux8 · 2024-08-12
> >
> > Thanks to the author for the response. After reading the response, most of my concerns have been addressed. I've decided to maintain my current score.

---

### Author Rebuttal · Authors · 2024-08-07

We supplemented Table 1 of the original manuscript with specific inference modes for each methods (@Reviewer-MJRN).

And we have also presented the visualization results on the CelebAMask-HQ dataset (@Reviewer-xcUS).

Please refer to the attached pdf.

---

### Decision · Program_Chairs · 2024-09-25

**Decision:**

Accept (poster)

**Comment:**

According to 3 out of 4 reviewers, this paper makes a solid contribution to the area of segmenting ultra highres images, and presents convincing results to show improvement over the state-of-the-art. One reviewer is an outlier, with a low score (3) and a very very short review. The authors alledge that review was written by an LLM. While the Area Chair was not able to validate that claim, the review is rather unprofessional and written in vague terms, well below the level of detail of the other reviews. Therefore the Area Chair largely discounted it and recommends this paper for acceptance.